# Phloroglucinol Inhibits Oxidative-Stress-Induced Cytotoxicity in C2C12 Murine Myoblasts through Nrf-2-Mediated Activation of HO-1

**DOI:** 10.3390/ijms24054637

**Published:** 2023-02-27

**Authors:** Cheol Park, Hee-Jae Cha, Hyun Hwangbo, Seon Yeong Ji, Da Hye Kim, Min Yeong Kim, EunJin Bang, Su Hyun Hong, Sung Ok Kim, Soon-Jeong Jeong, Hyesook Lee, Sung-Kwon Moon, Jung-Hyun Shim, Gi-Young Kim, Suengmok Cho, Yung Hyun Choi

**Affiliations:** 1Division of Basic Sciences, College of Liberal Studies, Dong-eui University, Busan 47340, Republic of Korea; 2Department of Parasitology and Genetics, College of Medicine, Kosin University, Busan 49267, Republic of Korea; 3Anti-Aging Research Center, Dong-eui University, Busan 47340, Republic of Korea; 4Department of Biochemistry, College of Korean Medicine, Dong-eui University, Busan 47227, Republic of Korea; 5Department of Food and Nutrition, College of Life and Health, Kyungsung University, Busan 48434, Republic of Korea; 6Department of Dental Hygiene & Institute of Basic Science for Well-Aging, Youngsan University, Yangsan 50510, Republic of Korea; 7Department of Convergence Medicine, Pusan National University School of Medicine, Yangsan 50612, Republic of Korea; 8Department of Food and Nutrition, College of Biotechnology & Natural Resource, Chung-Ang University, Ansung 17546, Republic of Korea; 9Department of Pharmacy, College of Pharmacy, Mokpo National University, Muan 58554, Republic of Korea; 10Department of Marine Life Science, College of Ocean Sciences, Jeju National University, Jeju 63243, Republic of Korea; 11Department of Food Science and Technology, Institute of Food Science, Pukyong National University, Busan 48513, Republic of Korea

**Keywords:** phloroglucinol, ROS, DNA damage, apoptosis, Nrf2/HO-1

## Abstract

Phloroglucinol is a class of polyphenolic compounds containing aromatic phenyl rings and is known to have various pharmacological activities. Recently, we reported that this compound isolated from *Ecklonia cava*, a brown alga belonging to the family *Laminariaceae*, has potent antioxidant activity in human dermal keratinocytes. In this study, we evaluated whether phloroglucinol could protect against hydrogen peroxide (H_2_O_2_)-induced oxidative damage in murine-derived C2C12 myoblasts. Our results revealed that phloroglucinol suppressed H_2_O_2_-induced cytotoxicity and DNA damage while blocking the production of reactive oxygen species. We also found that phloroglucinol protected cells from the induction of apoptosis associated with mitochondrial impairment caused by H_2_O_2_ treatment. Furthermore, phloroglucinol enhanced the phosphorylation of nuclear factor-erythroid-2 related factor 2 (Nrf2) as well as the expression and activity of heme oxygenase-1 (HO-1). However, such anti-apoptotic and cytoprotective effects of phloroglucinol were greatly abolished by the HO-1 inhibitor, suggesting that phloroglucinol could increase the Nrf2-mediated activity of HO-1 to protect C2C12 myoblasts from oxidative stress. Taken together, our results indicate that phloroglucinol has a strong antioxidant activity as an Nrf2 activator and may have therapeutic benefits for oxidative-stress-mediated muscle disease.

## 1. Introduction

Marine resources are receiving great attention as sources of new drug development. Among them, seaweed contains abundant natural products that can suppress oxidative-stress-dependent pathological conditions, including liver damage, diabetes, neurodegenerative diseases, muscle atrophy, cognitive impairment, dyslipidemia, and atherosclerosis [1,2,3]. In particular, studies on seaweed-derived phenolic compounds having excellent antioxidant activity are being actively conducted, and their antioxidant activities mainly involve the scavenging of reactive oxygen species (ROS) and activation of intracellular antioxidant signaling pathways [4,5,6]. Phloroglucinol, a polyphenol trihydroxybenzene bearing aromatic phenyl rings with three hydroxyl groups, is a naturally occurring secondary metabolite in various organisms, including marine brown algae [7,8,9]. Phloroglucinol is known to have various pharmacological potentials such as antibacterial, anticonvulsant, anti-allergic, antithrombotic, anti-inflammatory, and cancer chemopreventive activities [10,11,12]. Recently, the antioxidant potential of this polyphenolic compound was also confirmed in several in vitro and in vivo experimental models. For example, Drygalski et al. [13] reported that the strengthened antioxidant defense of phloroglucinol contributed to the reduction of hepatic steatosis and inflammatory response by reducing oxidative/nitrogen damage to cellular macromolecules. In addition, it was confirmed that phloroglucinol blocked the oxidative damage caused by hydrogen peroxide (H_2_O_2_) treatment or gamma-ray irradiation by affecting the activity of antioxidant and detoxifying enzymes in the retinal epithelium, hippocampal nerve, renal epithelial cells, and lung fibroblasts [14,15,16,17,18]. Moreover, phloroglucinol as a ROS scavenger modulates synaptic plasticity to attenuate the pathological phenomena of neurodegenerative diseases [19,20,21]. According to our previous study, phloroglucinol isolated from *Ecklonia cava* Kjellman of the *Laminariaceae* family was able to inhibit DNA damage and apoptosis in H_2_O_2_-exposed HaCaT human keratinocytes by the nuclear factor-erythrocyte 2-associated factor 2 (Nrf2)-dependent activation of heme oxygenase-1 (HO-1) [22]. Similar results have been confirmed in ultraviolet (UV) B-exposed keratinocytes [23], and the role of Nrf2 as an inhibitor of oxidative stress has also been emphasized in ROS-mediated osteoclastogenesis and Parkinson’s disease models [19,24]. However, the antioxidant effects and major signaling pathway of phloroglucinol in muscle are not fully demonstrated. Excessive ROS accumulation leads to deleterious effects, including a reduction in muscle force generation and induction of muscle atrophy. In addition, an increased ROS level may activate signaling pathways such as nuclear factor-kappa B associated with accelerated muscle aging [25,26]. Therefore, antioxidative phloroglucinol may contribute to reducing oxidative stress and treat aging-related muscle disease such as muscle atrophy.

Nrf2 is a master transcription factor that critically controls the transcriptional activity of several anti-oxidative phase II enzymes involved in the cellular defense against oxidative stress [27,28]. HO-1, a representative downstream regulator of Nrf2, is an iron-dependent cytoprotective enzyme whose metabolites contribute to the maintenance of redox homeostasis. Therefore, the Nrf2/HO-1 signaling pathway has received widespread attention as a major regulatory pathway of the intracellular defense against oxidative stress [28,29]. Although phloretin, a precursor of phloroglucinol, has been reported to exert antioxidant effects in muscle cells via the Nrf2 pathway [30,31], to date, the role of the Nrf2/HO-1 axis in the antioxidant activity of phloroglucinol is not fully understood. Therefore, the objective of this study was to evaluate the association between the antioxidant ability of phloroglucinol and the Nrf2/HO-1 signaling pathway against oxidative-stress-mediated cytotoxicity in muscle cells. For this purpose, an immortalized C2C12 murine myoblast model was used and H_2_O_2_ was treated to mimic oxidative stress.

## 2. Results

### 2.1. Phloroglucinol Inhibits the Decrease of Cell Viability Caused by H_2_O_2_ in C2C12 Cells

To evaluate the inhibitory effect of phloroglucinol against the H_2_O_2_-induced cytotoxicity on C2C12 myoblasts, cell viability was determined by performing a 3-(4,5-dimethylthiazol-2-yl)-2,5-diphenyltetra-zolium bromide (MTT) assay. As demonstrated in Figure 1A, cell viability was concentration-dependently reduced in C2C12 cells after H_2_O_2_. As the viability of cells treated with 1 mM H_2_O_2_ was inhibited by approximately 60%, 1 mM H_2_O_2_ was selected as the cytotoxicity-inducing concentration for all subsequent experiments. On the other hand, since phloroglucinol did not induce significant cytotoxicity at concentrations up to 20 μg/mL, this concentration was used as the optimal and highest concentration of phloroglucinol (Figure 1B). We then assessed the inhibitory effect of phloroglucinol on H_2_O_2_-mediated cytotoxicity and found that phloroglucinol significantly restored the H_2_O_2_-induced reduction of cell viability and morphological changes in thinned and contracted cells (Figure 1C,D), indicating that phloroglucinol pretreatment could improve H_2_O_2_-induced cytotoxicity.

### 2.2. Phloroglucinol Abolishes H_2_O_2_-Induced Apoptosis in C2C12 Cells

We next investigated whether phloroglucinol affected the induction of apoptosis due to H_2_O_2_ treatment. As presented in Figure 2A,B, the flow cytometry results after annexin V-fluorescein isothiocyanate (FITC)/propidium iodide (PI) staining revealed that much more apoptosis was induced in cells treated with H_2_O_2_ than in control cells. However, the induction of apoptosis by H_2_O_2_ was significantly protected by phloroglucinol pretreatment in a dose-dependent manner. In parallel with this, in H_2_O_2_-exposed C2C12 cells, DNA fragmentation and morphological changes characteristic of apoptosis, including nuclear fragmentation and chromatin condensation, were clearly detected, and these apoptotic features were strongly abrogated in phloroglucinol-pretreated cells (Figure 2C–E).

### 2.3. Phloroglucinol Protects H_2_O_2_-Induced Mitochondrial Impairment in C2C12 Cells

To investigate whether the inhibitory effect of phloroglucinol on the apoptosis induced by H_2_O_2_ was related to the protective ability of mitochondrial damage, we evaluated the effect of phloroglucinol on the H_2_O_2_-induced loss of mitochondrial membrane potential (MMP). Flow cytometry analysis results using 5,5′,6,6′-tetrachloro-1,1′3,3′-tetraethyl-imidacarbocyanune iodide (JC-1) staining showed that the frequency of JC-1 monomers was significantly increased whereas that of JC-1 aggregates was decreased in H_2_O_2_-treated cells, demonstrating the loss of MMP and consequent mitochondrial dysfunction (Figure 3A,B). Moreover, in H_2_O_2_-treated cells, the expression of cytochrome *c* was downregulated in the mitochondria but upregulated in the cytoplasm (Figure 3C). In addition, immunoblotting analysis revealed that H_2_O_2_ treatment induced a decrease in Bcl-2 expression, but there was no change in Bax expression, which was associated with the activation of caspase-3 and degradation of poly(ADP-ribose) polymerase (PARP) (Figure 3E). However, these changes were decreased in cells pretreated with phloroglucinol, suggesting that phloroglucinol was able to inhibit mitochondrial damage caused by H_2_O_2_.

### 2.4. Phloroglucinol Reduces H_2_O_2_-Induced DNA Damage and ROS Accumulation in C2C12 Cells

To determine whether the blocking ability of phloroglucinol against H_2_O_2_-mediated cytotoxicity is related to the protection of DNA damage, the effect of phloroglucinol on the H_2_O_2_-induced comet tail formation and phosphorylation level of γH2AX (p-γH2AX) was evaluated. As indicated in Figure 4A,B, a great increase in the comet tail moment (DNA migration) and expression of p-γH2AX were observed in H_2_O_2_-treated cells. However, the increase in these DNA damage marks was remarkedly weakened by phloroglucinol pretreatment, indicating that DNA damage caused by oxidative stress could be protected by phloroglucinol.

### 2.5. Phloroglucinol Diminishes H_2_O_2_-Induced ROS Accumulation and Activates Nrf2/HO-1 Signaling Pathway in C2C12 Cells

Because the mitochondria represent both the primary targets of ROS damage and major sources of ROS, we examined the effect of phloroglucinol on ROS formed by H_2_O_2_ using the cell-permeable fluorescent dye 2′,7′-dichlorofluorescein diacetate (DCF-DA). As can be seen in Figure 4C,D, the intensity of the average oxidized DCF peak was increased about 8.9-fold by H_2_O_2_ treatment compared to untreated control cells, and phloroglucinol pretreatment significantly attenuated the H_2_O_2_-induced ROS accumulation in a dose-dependent manner. Since Nrf2 is a potent master antioxidant transcriptional regulator, we subsequently explored whether the activation of Nrf2 was correlated with the antioxidant capacity of phloroglucinol. As shown in Figure 4E, the level of the phosphorylated form of Nrf2 (p-Nrf2) without changes in its total protein expression clearly increased with the increasing phloroglucinol treatment concentration, indicating that phloroglucinol acts as an Nrf2 activator. In addition, the expression and activity of HO-1, a representative downstream molecule of Nrf2, were also upregulated after phloroglucinol treatment (Figure 4E,G). Furthermore, although lower than that of phloroglucinol alone, the levels of p-Nrf2 and HO-1 expression as well as the activity of HO-1 were maintained to some extent in cells co-treated with H_2_O_2_ and phloroglucinol compared to control cells and cells treated with H_2_O_2_ alone (Figure 4F,G), suggesting that the Nrf2-mediated activation of HO-1 was increased by phloroglucinol in H_2_O_2_-treated C2C12 cells.

### 2.6. Activation of HO-1 by Phloroglucinol Contributes to Restoration of H_2_O_2_-Induced Oxidative Damage and Mitochondrial Dysfunction in C2C12 Cells

To evaluate whether the activity of HO-1 increased by phloroglucinol in H_2_O_2_-treated C2C12 cells was directly correlated with the antioxidant effect of phloroglucinol, we used a selective inhibitor of HO-1, zinc protoporphyrin IX (ZnPP). As shown in Figure 5A,B, our results indicated that the protective effect of phloroglucinol on ROS generation due to H_2_O_2_ treatment was effectively reversed by treatment with ZnPP. In parallel with this, pretreatment with ZnPP significantly attenuated the blocking effect of phloroglucinol on H_2_O_2_-induced MMP loss (Figure 5C,D). However, ZnPP alone did not affect the generation of ROS or change of MMP. Notably, the protective effects of phloroglucinol against the H_2_O_2_-induced cytoplasmic release of cytochrome *c*, decrease of the Bcl-2/Bax ratio, activation of caspase-3, and degradation of PARP were obviously reversed (Figure 5E–G). These findings suggest that phloroglucinol blocks H_2_O_2_-induced mitochondrial impairment in C2C12 cells through activating Nrf2-mediated HO-1.

### 2.7. Activation of HO-1 Is Involved in Mitigating H_2_O_2_-Induced Cytotoxicity by Phloroglucinol in C2C12 Cells

We further examined whether the activation of HO-1 was mediated by the DNA damage blocking effect of phloroglucinol in H_2_O_2_-treated C2C12 cells and found that the protective ability of phloroglucinol on the increased comet tail formation and p-γH2AX expression caused by H_2_O_2_ treatment was reversed in the presence of ZnPP (Figure 6A,B). Consistent with the results, we also found that pretreatment with ZnPP mitigated the preventive potential of phloroglucinol against H_2_O_2_-induced C2C12 cell apoptosis (Figure 6C–E). At the same time, pretreatment with ZnPP significantly abolished the alleviated effects of phloroglucinol against cytotoxicity in H_2_O_2_-exposed C2C12 cells (Figure 6F,G), demonstrating that HO-1 mediates cytoprotection against H_2_O_2_-mediated cytotoxicity.

## 3. Discussion

In the present study, we induced oxidative stress using H_2_O_2_ to investigate whether phloroglucinol could protect mouse-derived C2C12 myoblasts from oxidative injury and found that H_2_O_2_ induced mitochondrial dysfunction, DNA damage, and apoptotic cell death through an increase in ROS generation. However, phloroglucinol as an Nrf2 activator significantly blocked H_2_O_2_-induced cytotoxicity while scavenging ROS.

Mitochondrial and DNA damage caused by oxidative stimuli is mostly accompanied by the induction of apoptosis. Indeed, previous studies have demonstrated that the cytotoxic effect of H_2_O_2_ on C2C12 myoblasts is closely related to DNA damage and apoptosis [31,32,33,34]. The overload of ROS by oxidative stress can depolarize the mitochondrial membrane and contribute to the initiation of the mitochondria-mediated intrinsic apoptosis pathway [34,35]. This results in the loss of MMP, indicative of mitochondrial impairment, resulting in the release of cytochrome *c* to the cytoplasm from the mitochondria. In the cytoplasm, cytochrome *c* can activate the caspase cascade required for the intrinsic pathway, ultimately leading to the cleavage of PARP, a target protein of effector caspases including caspase-3, in the process of apoptosis [35,36]. Similar to previous results [31,37], the reduction of MMP, cytosolic release of cytochrome *c*, activation of caspase-3, and degradation of PARP were increased in H_2_O_2_-treated C2C12 myoblasts in the present study. However, there changes were greatly diminished by phloroglucinol pretreatment, and the inactivation of caspase-3 may have a causal relationship with the protection of apoptosis by H_2_O_2_.

Numerous accumulated studies have shown that the intrinsic apoptosis pathway is critically regulated by changes in the expression of Bcl-2 family proteins. Among them, anti-apoptotic proteins, including Bcl-2, are essential for maintaining the stability of the mitochondrial membrane barrier, whereas anti-apoptotic proteins such as Bax are key executors of mitochondrial poration [34,38]. Therefore, when the expression of Bax is relatively higher than that of Bcl-2, the mitochondrial membrane permeability is enhanced and the release of mitochondrial cytochrome *c* is promoted [36,38]. In our study, we found that the expression of Bcl-2 protein, which had been reduced by H_2_O_2_, was gradually restored to the control level as the phloroglucinol pretreatment concentration increased. Therefore, the restoration of the Bcl-2/Bax expression ratio after phloroglucinol pretreatment in H_2_O_2_-treated cells may have contributed to the blockade of MMP loss. In several previous studies, it has been reported that apoptosis, which is observed to be relatively high in myogenic progenitor cells derived from aged muscle, is responsible for the decrease in skeletal muscle regenerative capacity associated with high levels of ROS [39,40,41]. It was also found that the induction of mitochondria-mediated apoptosis observed in myoblasts exposed to oxidative stress was generally ROS-dependent [38,42,43]. In this study, H_2_O_2_ strongly increased the generation of ROS in C2C12 cells, but it was significantly blocked in the presence of phloroglucinol, and no significant production of ROS was detected by treatment with phloroglucinol alone. These findings suggest that phloroglucinol was able to protect C2C12 cells from oxidative-stress-induced DNA damage, mitochondrial dysfunction, and apoptosis while exerting a potent ROS scavenging activity.

Accumulating studies have identified several defense mechanisms linked to protecting cells from oxidative stress, which play key roles in maintaining cellular function by blocking the apoptosis pathway. Nrf2 is a transcription factor that can enhance the antioxidant capacity by promoting the expression of phase II detoxification enzymes [25,26]. As a defense mechanism against oxidative stress, Nrf2 regulates phase II detoxification enzymes and anti-oxidant enzymes, including NAD[P]H quinone oxidoreductase 1 and HO-1 [44]. When cells are exposed to Nrf2 activators, Nrf2 must be phosphorylated for nuclear translocation after the dissociation from Kelch-like ECH-associated protein 1 (Keap1) to increase the transcriptional activity of Nrf2-dependent antioxidant genes such as HO-1. This activates a series of events that provide protection against the oxidative challenge. Based on previous reports, phloroglucinol exerted beneficial antioxidative effects by activating Nrf2 activity, thus increasing levels of Nrf2 antioxidant enzymes such as catalase and glutathione peroxidase [19]. Furthermore, phloretin, a precursor of phloroglucinol, protected against oxidative stress through the extracellular signal-regulated kinase/Nrf2 signaling pathway [45]. In addition, as one of the representative downstream factors of Nrf2, HO-1 can decompose heme into free iron, biliverdin, and carbon monoxide, and the produced biliverdin is converted into bilirubin, which has a strong antioxidant activity [25,27]. These findings indicate that discovering substances capable of activating the Nrf2/HO-1 axis may be an appropriate strategy to counteract cellular damage caused by oxidative injury. Indeed, several previous studies have reported that the Nrf2-mediated activation of HO-1 in myoblasts may serve as a protective mechanism against apoptosis induced after mitochondrial dysfunction caused by oxidative stress [20,29]. Therefore, we evaluated whether phloroglucinol could activate Nrf2 in C2C12 myoblasts and found that phloroglucinol increased the phosphorylation of Nrf2. Phloroglucinol also upregulated the expression of HO-1 and its enzymatic activity, suggesting that phloroglucinol may act as an activator of Nrf2 to increase the expression of HO-1. Moreover, although the expression of p-Nrf2 and activation of HO-1 did not show a synergistic effect in cells treated with both H_2_O_2_ and phloroglucinol, the phosphorylation of Nrf2 and activity of HO-1 were further enhanced compared to cells treated with H_2_O_2_ alone or control cells. In further experiments with ZnPP, an HO-1 inhibitor, the mitochondrial protective potential and apoptosis-blocking ability by phloroglucinol in H_2_O_2_-treated cells were largely offset, as the antioxidant efficacy was counteracted. These data demonstrated that the Nrf2-mediated activation of HO-1 was at least responsible for the blockade of H_2_O_2_-induced oxidative damage by phloroglucinol. Our results support well the previous studies conducted in several in vitro and in vivo models showing that the antioxidant activity of phloroglucinol is due to the enhanced activity of the Nrf2/HO-1 signaling [19,22,23,24]. Consequently, our results indicate that the Nrf2-mediated activation of HO-1 by phloroglucinol served as an upstream signal for the blocking action of phloroglucinol against H_2_O_2_-induced cytotoxicity in C2C12 myoblasts.

## 4. Materials and Methods

### 4.1. Reagents and Materials

All materials necessary for cell culture were obtained from WELGENE Inc. (Gyeongsan, Republic of Korea). Phloroglucinol (1,3,5-trihydroxybenzene), H_2_O_2_, MTT, JC-1, mitochondrial fractionation kit, and enhanced chemiluminescence (ECL) solution were purchased from Thermo Fisher Scientific (Waltham, MA, USA). Dimethyl sulfoxide (DMSO), ZnPP, DCF-DA, DAPI, RNase A, proteinase K, isopropyl alcohol, EtBr, and caspase-3 activity assay kit were obtained from Sigma-Aldrich Co. (St. Louis, MO, USA). The comet assay kit was purchased from Trevigen, Inc. (Gaithersburg, MD, USA). HO-1 enzyme-linked immunosorbent assay (ELISA) kit and annexin V-FITC apoptosis detection kit were purchased from Abcam, Inc. (Cambridge, UK). Immobilon^®^-P PVDF membranes were obtained from Merck Millipore (Bedford, MA, USA). Primary and horseradish peroxidase-conjugated secondary antibodies were purchased from Cell Signaling Technology (Beverly, MA, USA), Santa Cruz Biotechnology, Inc. (Santa Cruz, CA, USA), and Abcam, Inc (Cambridge, UK).

### 4.2. Cell Culture and Treatment

C2C12 myoblasts purchased from the American Type Culture Collection (Manassas, VA, USA) were maintained according to a previously described method [46]. Stock solutions of phloroglucinol and H_2_O_2_ were prepared by dissolving them in DMSO. They were used to treat cells after dilution to appropriate concentrations with culture medium just before the experiment. To investigate the effect of phloroglucinol on oxidative damage following H_2_O_2_ treatment, C2C12 cells were cultured in media containing various concentrations of phloroglucinol and H_2_O_2_ for 24 h or pretreated with phloroglucinol and/or ZnPP for 1 h before they were treated with H_2_O_2_ for 24 h. To evaluate the blocking activity of phloroglucinol on the generation of ROS by H_2_O_2_, C2C12 cells were stimulated with phloroglucinol and/or ZnPP for 1 h before they were treated with H_2_O_2_ for 1 h.

### 4.3. Cell Viability Assay and Cell Morphological Change Observation

The cell viability of C2C12 cells cultured under various treatment conditions was investigated through MTT assay using the same method as previously described [47]. After treatment, the morphological changes of cells were captured under a phase-contrast microscope (Carl Zeiss, Oberkochen, Germany).

### 4.4. Flow Cytometry Analysis

For quantitative evaluation of apoptosis-induced cells, the collected cells were fixed and stained with annexin V-FITC and PI. Subsequently, annexin V-positive cells were considered as cells in which apoptosis was induced through flow cytometry analysis as described previously [46]. To determine the levels of MMP, cells were stained with JC-1. The percentage of JC-1 monomers using a flow cytometer was expressed to indicate cells that lost MMP [44,48]. For quantitative evaluation of ROS generation by DCF-DA staining, intensities of DCF fluorescence reflecting ROS generation were detected by flow cytometry as described previously [49].

### 4.5. Apoptosis Analysis via DAPI Staining

After collecting cells to monitor apoptosis through DAPI staining, the cells were fixed with 4% paraformaldehyde according to the preceding method [50]. The cells were subjected to staining with DAPI solution and morphological changes of the nuclei were taken using a fluorescence microscope (Carl Zeiss) at Core-Facility Center for Tissue Regeneration, Dong-eui University (Busan, Korea).

### 4.6. DNA Fragmentation Assay

In order to observe fragmented DNA, which is an apoptosis marker, the cell pellet was suspended in a lysis solution as described previously [51]. The supernatants were incubated with RNase A and proteinase K, and DNA was precipitated with isopropyl alcohol. The extracted DNA was fractionated using 1.0% agarose gel. Subsequent to electrophoresis, the gel was stained with EtBr and the fragmentation pattern was visualized under UV light.

### 4.7. Protein Isolation and Immunoblotting

Total protein for immunoblotting was isolated, extracted as previously described [52]. Cytoplasmic and mitochondrial fractions were extracted using a mitochondrial fractionation kit following the manufacturer’s protocol. After protein quantification, the same amount of protein was separated using sodium dodecyl sulfate-polyacrylamide gels and transferred to membranes. After blocking membranes with 5% non-fat dry milk, they were incubated with primary antibodies and then reacted with secondary antibodies. Membrane-bound antibodies were visualized with ECL solution. Actin was used as an internal control for total and cytoplasmic proteins. Cytochrome oxidase subunit 4 (COX IV) was used as a loading marker for mitochondrial proteins.

### 4.8. Comet Assay

DNA damage analysis was detected using a Comet assay kit following the manufacturer’s instructions. In brief, cells treated with H_2_O_2_ in the presence or absence of phloroglucinol and/or ZnPP were suspended in 1% low melting point agarose and spread on comet slides. After DNA denaturation, electrophoresis was performed and stained with an asymmetrical cyanine dye. Fluorescence images were observed and captured using a fluorescence microscope.

### 4.9. Analysis of Caspase-3 Activity

Caspase-3 activity was quantified using the fluorescent substrate of caspase-3, acetyl-Asp-Glu-Val-Asp-chromophore-p-nitroanilide (Ac-DVAD-pNa), according to the manufacturer’s instructions. Enzyme-catalyzed release of pNa was monitored at 405 nm using a microplate reader and the activity of caspase-3 was presented relative to the control [53].

### 4.10. Analysis of HO-1 Activity

To measure the activity of HO-1, the amount of bilirubin formed in the heme of cells cultured under various conditions was evaluated using an HO-1 ELISA kit. The levels of bilirubin were calculated based on the difference in absorption at 510 nm according to the method suggested by the manufacturer. As previously described [54], the activity of HO-1 was expressed as fold change relative to the control.

### 4.11. Statistical Analysis

Data are expressed as mean ± standard deviation (SD). Student’s *t*-test using GraphPad Prism (Version 5.0) (Graphpad Inc., San Diego, CA, USA) was employed for statistical analyses. *p* < 0.05 was considered to indicate a statistically significant difference (* *p* ˂ 0.05 and *** *p* ˂ 0.001 vs. control group; ^#^ *p* ˂ 0.05 and ^###^ *p* ˂ 0.001 vs. H_2_O_2_-treated cells; ^$$^ *p* ˂ 0.01 and ^$$$^ *p* ˂ 0.001 vs. with phloroglucinol + H_2_O_2_ treatment group).

## 5. Conclusions

In summary, our results indicated that phloroglucinol could alleviate DNA damage and apoptosis by mitigating H_2_O_2_-induced mitochondrial damage as an ROS scavenger in C2C12 myoblasts. In addition, phloroglucinol as an activator of Nrf2 might contribute to blocking oxidative damage by promoting the expression of HO-1, which is thought to be because the enhanced ROS production by H_2_O_2_ was eliminated by the activation of HO-1 (Figure 7). Although further experiments are needed to better understand the molecular mechanisms involved in the activation of Nrf2, our findings confirm the protective role of phloroglucinol in oxidative-stress-related skeletal muscle disease. Nevertheless, it is also necessary to explore other intracellular signaling pathways that may be involved in phloroglucinol-mediated antioxidant activity and to confirm their efficacy in animal models. In addition, the antioxidative role of phloroglucinol needs to be further investigated in terminally differentiated myotubes under oxidative stress during skeletal muscle physiology and pathology.

## Figures and Tables

**Figure 1 ijms-24-04637-f001:**
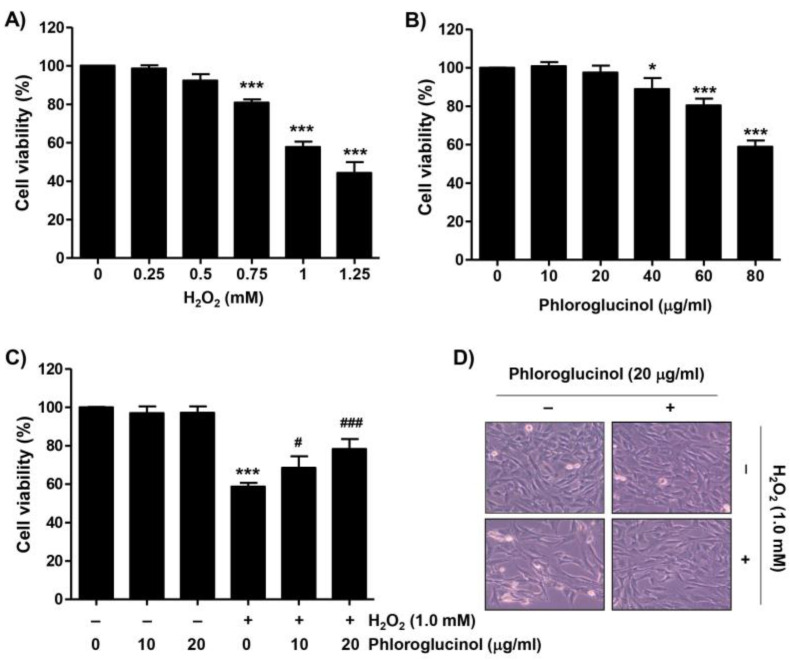
Phloroglucinol protects H_2_O_2_-induced decrease of cell viability in C2C12 myoblasts. (**A**–**C**) 3-(4,5-dimethylthiazol-2-yl)-2,5-diphenyltetra-zolium bromide assay was performed after treatment of cells with the indicated concentrations of H_2_O_2_ or vehicle for control group (**A**) or phloroglucinol (**B**) for 24 h or pretreatment for 1 h with or without phloroglucinol followed by stimulation with H_2_O_2_ for an additional 24 h (**C**,**D**). Representative morphological images of cells were captured under an inverted-phase contrast microscope (200×). Statistical difference was determined by Student’s *t*-test (* *p* ˂ 0.05 and *** *p* ˂ 0.001 vs. control group; ^#^
*p* ˂ 0.05 and ^###^
*p* ˂ 0.001 vs. H_2_O_2_-treated cells).

**Figure 2 ijms-24-04637-f002:**
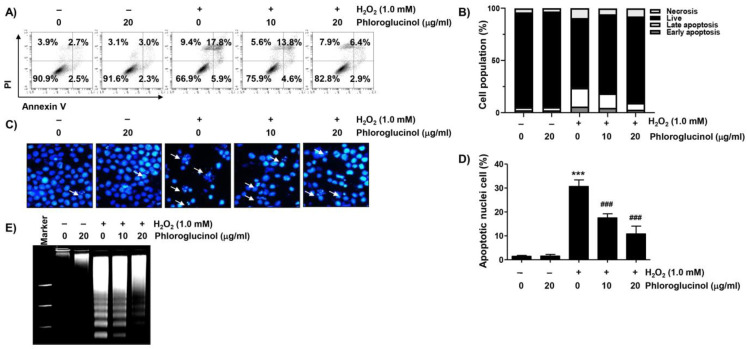
Phloroglucinol ameliorates cells against apoptosis in H_2_O_2_-treated C2C12 myoblasts. Cells were exposed to phloroglucinol for 1 h prior to treatment with H_2_O_2_ for 24 h. Cells were exposed to vehicle for control group. (**A**,**B**) To measure the frequency of apoptosis, flow cytometry was performed after staining with annexin V-fluorescein isothiocyanate (FITC) and propidium iodide (PI). (**A**) Representative histograms. Cells were classified as healthy cells (lower left quadrant, annexin V^−^/PI^−^), early apoptotic cells (lower right quadrant, annexin V^+^/PI^−^), late apoptotic cells (upper right quadrant, annexin V^+^/PI^+^), and necrotic cells (upper left quadrant, annexin V^−^/PI^+^). (**B**) Quantitative results following flow cytometry are presented. (**C**) After 4,6-diamidino-2-phenolindole (DAPI) staining, images of representative nuclei are presented (400×). Arrows demonstrate apoptotic cells in different groups. (**D**) The frequency of apoptotic cells per slide was estimated by counting apoptotic cells. (**E**) Isolated genomic DNA was subjected to agarose gel electrophoresis. The DNA was visualized with ethidium bromide (EtBr) staining. Statistical difference was determined by Student’s *t*-test (*** *p* ˂ 0.001 vs. control group; ^###^
*p* ˂ 0.001 vs. H_2_O_2_-treated cells).

**Figure 3 ijms-24-04637-f003:**
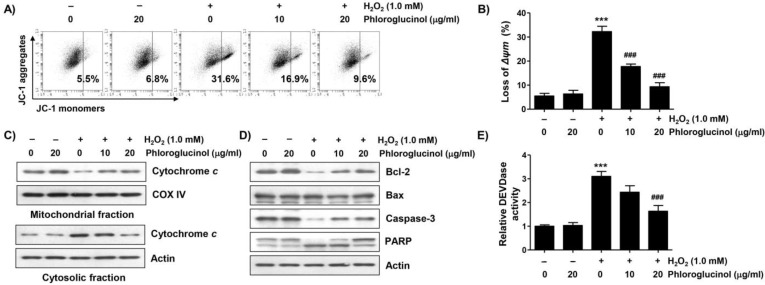
Phloroglucinol suppresses H_2_O_2_-induced mitochondrial impairment in C2C12 myoblasts. Cells treated with or without phloroglucinol for 1 h were treated with H_2_O_2_ for 24 h. Cells were exposed to vehicle for control group. (**A**) Representative flow cytometry results according to 5,5′,6,6′-tetrachloro-1,1′3,3′-tetraethyl-imidacarbocyanune iodide (JC-1) staining are presented. (**B**) The JC-1 monomer ratio of cells in each treatment group was presented. (**C**,**D**) The expression levels of cytochrome *c* (**C**), Bcl-2, Bax, caspase-3, and poly(ADP-ribose) polymerase (PARP) (**D**) were investigated through immunoblotting. (**E**) Caspase-3 activity was measured using a colorimetric substrate. Statistical difference was determined by Student’s *t*-test (*** *p* ˂ 0.001 vs. control group; ^###^
*p* ˂ 0.001 vs. H_2_O_2_-treated cells).

**Figure 4 ijms-24-04637-f004:**
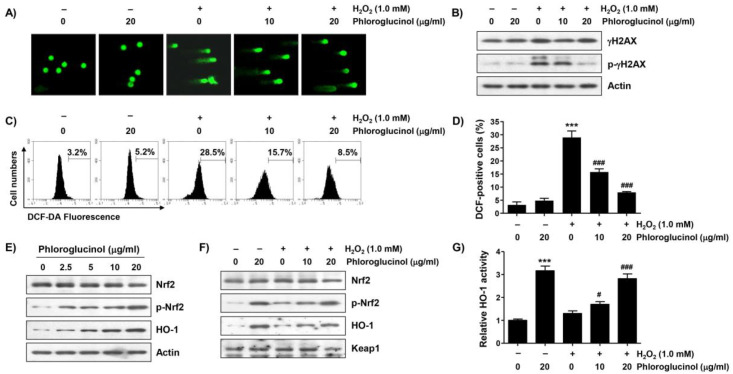
Phloroglucinol attenuates H_2_O_2_-induced DNA damage and ROS production and activates nuclear factor-erythroid-2 related factor 2 (Nrf2) in C2C12 myoblasts. Cells exposed with or without phloroglucinol for 1 h were stimulated with H_2_O_2_ for another 24 h (**A**,**B**,**F**,**G**) or 1 h (**C**,**D**) or cells were incubated for 24 h in a medium containing the indicated concentrations of phloroglucinol (**E**). Cells were exposed to vehicle for control group. (**A**) Representative images following comet assay are shown (400×). (**B**) Changes in expression of γH2AX and its phosphorylated form (p-γH2AX) were determined by immunoblotting. (**C**,**D**) The levels of ROS production were investigated by performing 2′,7′-dichlorofluorescein diacetate (DCF-DA) staining, and representative histograms (**C**) and results (**D**) of flow cytometry are presented. (**E**,**F**) After extracting cell lysate for each treatment group, expression levels of presented proteins were investigated through immunoblotting. (**G**) The activity of heme oxygenase-1 (HO-1) is presented as a relative value. Statistical difference was determined by Student’s *t*-test (*** *p* ˂ 0.001 vs. control group; ^#^
*p* ˂ 0.05 and ^###^
*p* ˂ 0.001 vs. H_2_O_2_-treated cells).

**Figure 5 ijms-24-04637-f005:**
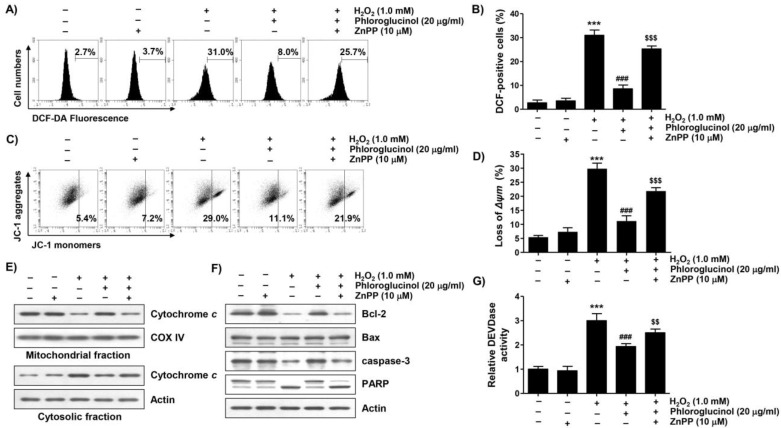
Zinc protoporphyrin IX (ZnPP) abrogates the inhibitory effect of phloroglucinol on ROS generation and mitochondrial damage in H_2_O_2_-treated C2C12 myoblasts. Cells treated with phloroglucinol and/or ZnPP for 1 h were treated with H_2_O_2_ for 1 h (**A**,**B**) or 24 h (**C**–**G**). Cells were exposed to vehicle for control group. (**A**,**B**) Representative flow cytometry results according to DCF-DA staining (**A**) and their average values (**B**) are presented. (**C**) Representative flow cytometry results according to JC-1 staining are presented. (**D**) The JC-1 monomer ratio of cells in each treatment group was shown. (**E**,**F**) The levels of cytochrome *c* (**E**), Bcl-2, Bax, caspase-3, and PARP expression (**F**) were investigated through immunoblotting. (**G**) Caspase-3 activity was measured using a colorimetric substrate. Statistical difference was determined by Student’s *t*-test (*** *p* ˂ 0.001 vs. control group; ^###^ *p* ˂ 0.001 vs. H_2_O_2_-treated cells; ^$$^ *p* ˂ 0.01, ^$$$^ *p* ˂ 0.001 vs. phloroglucinol + H_2_O_2_ treatment group).

**Figure 6 ijms-24-04637-f006:**
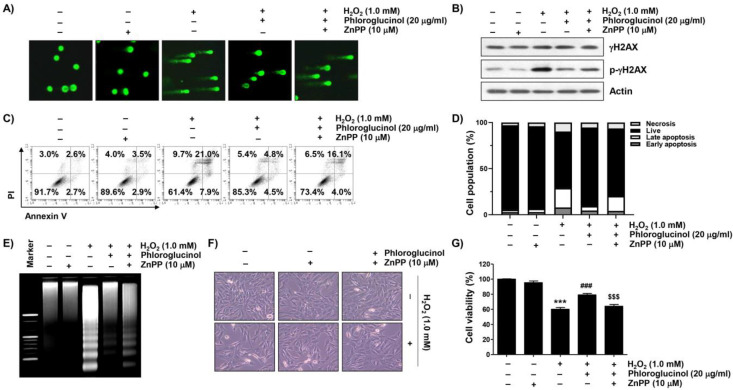
The inhibitory ability of phloroglucinol on DNA damage and apoptosis-induced by H_2_O_2_ is offset by ZnPP in C2C12 myoblasts. Cells were treated with phloroglucinol and/or ZnPP for 1 h and then treated with H_2_O_2_ for 24 h. Cells were exposed to vehicle for control group. (**A**) Representative images were captured after performing comet analysis (400×). (**B**) Changes in expression of γH2AX and its phosphorylated form (p-γH2AX) were determined by immunoblotting. (**C**,**D**) To measure the frequency of apoptosis, flow cytometry was performed after double staining with annexin V and PI. (**C**) Cells were classified as healthy cells (lower left quadrant, annexin V^−^/PI^−^), early apoptotic cells (lower right quadrant, annexin V^+^/PI^−^), late apoptotic cells (upper right quadrant, annexin V^+^/PI^+^), and necrotic cells (upper left quadrant, annexin V^−^/PI^+^). (**D**) Quantitative results following flow cytometry are presented. (**E**) Genomic DNA was visualized by EtBr staining after electrophoresis on an agarose gel. (**F**) Representative morphological images were indicated. (**G**) Cell viability was assessed by MTT assay. Statistical difference was determined by Student’s *t*-test (*** *p* ˂ 0.001 vs. control group; ^###^ *p* ˂ 0.001 vs. H_2_O_2_-treated cells; ^$$$^ *p* ˂ 0.001 vs. phloroglucinol + H_2_O_2_ treatment group).

**Figure 7 ijms-24-04637-f007:**
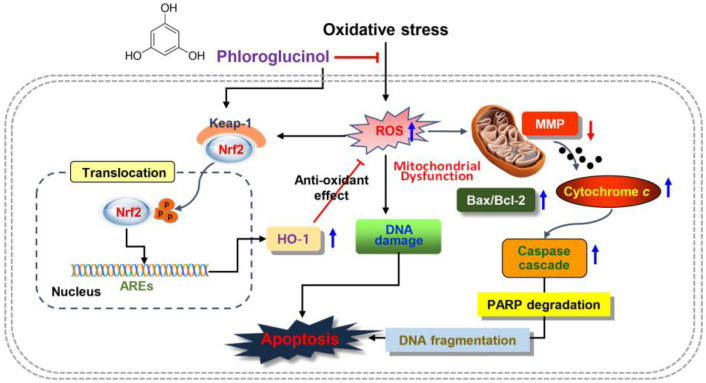
Schematic of the role of phloroglucinol on oxidative injury in C2C12 myoblasts. Phloroglucinol as an activator of Nrf2 signaling and scavenger of ROS protected cells by blocking H_2_O_2_-induced DNA and mitochondrial damage and apoptosis.

## Data Availability

The data presented in this study are available upon request from the corresponding author.

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
