# Peer review of "Phloroglucinol Inhibits Oxidative-Stress-Induced Cytotoxicity in C2C12 Murine Myoblasts through Nrf-2-Mediated Activation of HO-1"

_ijms, 2023, doi:10.3390/ijms24054637_

Round 1
Reviewer 1 Report
In their work, the authors presented interesting results confirming the cytoprotective role of phloroglucinol which affected the DNA damage and apoptosis level, working as a ROS scavenger in skeletal muscles against the H2O2-induced oxidative damage. The tested compound activated the Nrf2 signaling pathway, which might contribute to its antioxidant properties. The authors performed comprehensive and valid analyses. However, before the work is published, I would like the authors to address the following comments:
1. The introduction should describe the applicative character of phloroglucinol in the oxidative stress-mediated muscle diseases mentioned in the abstract and not described in the introduction. What was the reason for using immortalized murine muscle cells in this research model?
2. I would also like the authors to emphasize what type of activation of the Nrf2 signaling pathway is affected by the compound they examined. In addition, it should be noted that Nrf2 activates the expression of antioxidant and detoxification genes. Cellular biotransformation of xenobiotics is divided into phases 1 and 2. Phase II enzymes are major detoxification enzymes.
3. Lines 86-87 states that the role of Nrf2/OH-1 is very limited. Was that the intended meaning?
4. In line 101, there is a 20 mg/ml concentration, and the described one is 20 ug/ml.
5. The figures' captions lack statistical data and information about the controls.
6. Are the microscope magnifications correctly indicated (200x and 400x)?
7. DNA damage from DAPI-stained cells should be presented and described more clearly.
8. Flow cytometry results should also indicate if the results and distribution of cells in gates come from early/late apoptotic cells or dead cells, same as graphs presenting the percentages of apoptotic cells.
After introducing the proposed changes, I will be pleased to recommend the work for publication in the International Journal of Molecular Sciences.
Author Response
Reviewer 1:
In their work, the authors presented interesting results confirming the cytoprotective role of phloroglucinol which affected the DNA damage and apoptosis level, working as a ROS scavenger in skeletal muscles against the H2O2-induced oxidative damage. The tested compound activated the Nrf2 signaling pathway, which might contribute to its antioxidant properties. The authors performed comprehensive and valid analyses. However, before the work is published, I would like the authors to address the following comments:
Response: We highly appreciate you for providing us careful reviews and critical comments. Based on your review comments, we put our best efforts to adequately address all critical questions point-to-point in this response letter. We hope that these responses and changes will substantially improve our manuscript and facilitate your decision to publish study in your journal. We sincerely appreciate for your dedicated comments.
1. The introduction should describe the applicative character of phloroglucinol in the oxidative stress-mediated muscle diseases mentioned in the abstract and not described in the introduction. What was the reason for using immortalized murine muscle cells in this research model?
Response: Thank you for your highly productive recommendation. We fully agree with your comment. Therefore, we added description on potential application of antioxidative phloroglucinol in oxidative stress-mediated muscle diseases (line 78, page 2). And we used immortalized murine muscle cell line for our research purpose due to numbers of advantages. These immortalized cell lines are relatively convenient to conduct in vitro experiments as it provides an unlimited supply of cells due to its immortal nature. Additionally, we used established muscle cell line instead of primary cultured cells as it bypasses ethical issues associated with the use of animal tissue. We hope that this answers your question.
2. I would also like the authors to emphasize what type of activation of the Nrf2 signaling pathway is affected by the compound they examined. In addition, it should be noted that Nrf2 activates the expression of antioxidant and detoxification genes. Cellular biotransformation of xenobiotics is divided into phases 1 and 2. Phase II enzymes are major detoxification enzymes.
Response: We appreciate for your highly critical comment. Based on our data, we demonstrated that phloroglucinol activates nuclear factor erythroid-2-related (Nrf2) as main regulator. Furthermore, we conducted literature search and found that other signaling pathways that are activated along with Nrf2. Please see line 319, page 9 for the description. We also noted that Nrf2 activates the expression of both antioxidant and detoxification genes for its antioxidative mechanism and we described them in the Discussion section (line 312, page 9). We highly appreciate for your important comments.
3. Lines 86-87 states that the role of Nrf2/OH-1 is very limited. Was that the intended meaning?
Response: Thank you for pointing out ambiguous sentence. In order to avoid confusion, we changed the wording “very limited” to “not fully understood” to convey that anti-oxidative effect of phloroglucinol on muscle cells through Nrf2/HO-1 signaling pathway is not fully known. We hope that this provides clarity. Thank you for your suggestion.
4. In line 101, there is a 20 mg/ml concentration, and the described one is 20 ug/ml.
Response: We highly appreciate for indicating this unit error. It was a typo and we corrected “mg/ml” to “μg/ml”.
5. The figures' captions lack statistical data and information about the controls.
Response: Thank you for your precise and detailed comment. As per your recommendation, we newly added statistical information to each Figures legends and information about the control group. Thank you for your very important suggestions.
6. Are the microscope magnifications correctly indicated (200x and 400x)?
Response: Thank you for your comment. Yes, we correctly indicated the microscope magnifications for 200x and 400x. Thank you again for your detailed question.
7. DNA damage from DAPI-stained cells should be presented and described more clearly.
Response: We highly appreciate for your precise comment. Following your recommendation, we re-selected microscopic images that would better represent and describe our data and interpretation. Please find newly added DAPI-stained cell images with white arrows pointing towards apoptotic nuclei cells in Figure 2C. Additionally we quantified apoptotic nuclei cells (%) in Figure 2D, respectively. We thank you again for your suggestions that could improve our manuscript.
8. Flow cytometry results should also indicate if the results and distribution of cells in gates come from early/late apoptotic cells or dead cells, same as graphs presenting the percentages of apoptotic cells. After introducing the proposed changes, I will be pleased to recommend the work for publication in the International Journal of Molecular Sciences.
Response: Thank you for bringing up important point. We agree with you that results should be indicated with early and late apoptotic cells and live and necrotic cells. Following your recommendation, we revised Figure 2A, 2B, 6C, and 6D. Thank you again for your productive suggestion for further improving our manuscript.
Reviewer 2 Report
In this manuscript, authors highlighted the capacity of phloroglucinol to alleviate DNA damage and apoptosis by mitigating H2O2-induced mitochondrial damage as an ROS scavenger in C2C12 myoblasts. In addition, they fond that phloroglucinol, as an activator of Nrf2, might contribute to blocking oxidative damage by promoting the expression of HO-1.
This article is well written and structured from an experimental point of view.
There are just few my minor comments to address:
· Page 10, line 405. I would suggest to move the "Conclusions" after the "Discussion";
· In the Conclusions suction, it is also important to state the possible extension of this study on muscle cells terminally differentiated (myotubes). Since our muscle fibers can be exposed to ROS during physiological and pathological condition, it would be interesting to see the possible “beneficial effects” of phloroglucinol.
Author Response
Reviewer 2:
In this manuscript, authors highlighted the capacity of phloroglucinol to alleviate DNA damage and apoptosis by mitigating H2O2-induced mitochondrial damage as an ROS scavenger in C2C12 myoblasts. In addition, they found that phloroglucinol, as an activator of Nrf2, might contribute to blocking oxidative damage by promoting the expression of HO-1.
This article is well written and structured from an experimental point of view.
Response: We highly appreciate for your critical reviews and comments. While incorporating our responses to your comments, it was very helpful for us to revise our manuscript. We truly hope that our point-to-point responses below could resolve your inquiries and substantially improve our manuscript. We highly appreciate you for this opportunity to revise our manuscript.
1. Page 10, line 405. I would suggest to move the "Conclusions" after the "Discussion";
Response: Thank you for recommendation on the manuscript structure. Kindly note that our current manuscript structure is based on the authors guideline provided by the International Journal of Molecular Sciences Journal. Therefore, we have to abide with the Journal policy. Therefore, we were not to move “Conclusions” after the “Discussion”. Thank you again for your constructive suggestion.
2. In the Conclusions suction, it is also important to state the possible extension of this study on muscle cells terminally differentiated (myotubes). Since our muscle fibers can be exposed to ROS during physiological and pathological condition, it would be interesting to see the possible “beneficial effects” of phloroglucinol.
Response: We highly appreciate for your insightful comment. We newly described your recommended future study in our Discussion Section (line 451, page 12). Thank you for your productive suggestion.